# 10-year survival outcome after clinically suspected acute myocarditis in adults: A nationwide study in the pre-COVID-19 era

**Mi-Jeong Kim[1,2], Hae Ok Jung[2,3]\*, Hoseob Kim[4], Yoonjong Bae[4], So Young Lee[3], Doo Soo Jeon[1,2]**

**1** Division of Cardiology, Department of Internal Medicine, Incheon St. Mary's Hospital, The Catholic University of Korea, Incheon, Republic of Korea, **2** Catholic Research Institute for Intractable Cardiovascular Disease, College of Medicine, The Catholic University of Korea, Seoul, Republic of Korea, **3** Division of Cardiology, Department of Internal Medicine, Seoul St. Mary's Hospital, The Catholic University of Korea, Seoul, Republic of Korea, **4** Department of Data Science, Hanmi Pharm. Co., Ltd., Seoul, Republic of Korea

\* hojheart@catholic.ac.kr

## Abstract

### Background

Clinical courses of acute myocarditis are heterogeneous in populations and geographic regions. There is a dearth of long-term outcomes data for acute myocarditis prior to the coronavirus disease pandemic, particularly in the older and female population. This study aimed to provide the nationwide epidemiologic approximates of clinically suspected acute myocarditis across adults of all ages over the long term.

### Methods

From the nationwide governmental health insurance database, a retrospective cohort comprised all patients aged 20–79 who were hospitalized for clinically suspected acute myocarditis without underlying cardiac diseases from 2006 to 2018. The complicated phenotype was defined as requiring hemodynamic or major organ support. Over 10 years, all-cause mortality and index event-driven excess mortality were evaluated according to young-adult (20–39 years), mid-life (40–59 years), and older-adult (60–79 years) age groups.

### Results

Among 2,988 patients (51.0±16.9 years, 46.2% women), 362 (12.1%) were of complicated phenotype. Of these, 163 (45.0%) had died within 1 month. All-cause death at 30 days occurred in 40 (4.7%), 52 (4.8%), and 105 (10.0%) patients in the young-adult, mid-life, and older-adult groups, respectively. For 10 years of follow-up, all-cause death occurred in 762 (25.5%). Even in young adult patients with non-complicated phenotypes, excess mortality remained higher compared to the general population.

**Data Availability Statement:** This study used data provided by the National Health Insurance Service (NHIS-2021-1-338). The NHIS database is available to researchers who meet the criteria for

access. Requests should be directed to: https://nhiss.nhis.or.kr/bd/ab/bdaba041eng.do.

**Funding:** This work was supported by the Research Fund of Seoul St. Mary's Hospital, The Catholic University of Korea. The funders had no role in study design, data collection and analysis, decision to publish, or preparation of the manuscript.

**Competing interests:** I have read the journal's policy and the authors of this manuscript have the following competing interests: Hoseob Kim and Yoonjong Bae work at Hanmi Pharm. Co., Ltd, and performed blinded statistical analyses, based on industry-university cooperation, and there is no conflict of interest as a third party in this study.

## Conclusion

In hospitalized patients with clinically suspected acute myocarditis, short-term mortality is high both in young and older adults, particularly those with comorbidities and severe clinical presentation. Furthermore, excess mortality remains high for at least 10 years after index hospitalization in young adults.

## Introduction

Acute myocarditis is a heterogenous myocardial inflammatory syndrome presenting through various etiologies, clinical presentations, and prognoses [1]. As viral infection is the most frequent cause, acute myocarditis varies in endemic features depending on the causative microorganisms [2]. Autoimmune myocarditis and cardiotoxic chemical-related myocardial injury are known to be the rest of the poorly defined causes [3]. In Korea, acute myocarditis was reported in 1.4% of patients admitted to a tertiary hospital for acute heart failure before the start of the COVID-19 pandemic, however, detailed epidemiological data focused on acute myocarditis itself are not yet available [4].

Acute myocarditis has often been featured in the younger population, however, the COVID-19 pandemic clearly reveals that middle-aged and older populations also experience viral myocarditis, with more serious consequences than in younger people [5, 6]. There are little data on middle-aged individuals and older adults. In addition, current evidence shows that the long-term health impacts of acute myocarditis vary depending on the study population [7–22]. We aimed to estimate the nationwide epidemiologic approximates of clinically suspected acute myocarditis across adults of all ages in the pre-COVID-19 era, and to provide relevant data on acute myocarditis-associated mortalities over the long term.

## Methods

### Study population

We conducted a retrospective nationwide cohort study using the national registry from the National Health Insurance Service (NHIS) database, which is an obligatory public medical insurance system operated by the Korean government and covers the country's entire population [23]. Because this database primarily included the information used for insurance claims, we designed the operational definition to include and exclude patients with clinically suspected acute myocarditis [24, 25]. Inclusion criterion was all patients aged 20–79 years who were hospitalized for ≥5 days due to the clinically suspected first onset acute myocarditis from January 2006 to December 2018. First, patients with a primary diagnosis of acute myocarditis were identified by the International Classification of Disease (Tenth Revision; ICD-10) codes of I01.2, I09.0, I40.0–40.9, I41.0–41.8, and I51.4. Then, anyone with an alternative diagnosis other than the first onset de novo acute myocarditis was excluded. The exclusion criteria were any pre-existing conditions of acute or chronic ischemic heart diseases, heart failure, cardiomyopathies, valvular heart diseases, infective endocarditis, and genetic arrhythmic disorders. Pregnancy and cardiac surgery within 3 months before and after the index hospitalization were also excluded. In addition, systemic disorders including thyrotoxicosis, alcoholic cardiomyopathy, Chagas disease, toxoplasmosis, and parasite-related myocarditis were excluded. These pre-specified exclusion criteria were confirmed using all the registered national insurance codes to claim the expenses for drugs, diagnostic and therapeutic procedures, and surgeries as well as the ICD codes for diagnosis.

As the characteristics of acute myocarditis are considered to be potentially affected by age, the study population was classified into three groups according to age: young adult (20–39 years), mid-life adult (40–59 years), and older adult (60–79 years).

The study conformed with the tenets of the Declaration of Helsinki. The study protocol was approved by the Institutional Review Boards with an exemption (KIRB-E20200227-001). Informed consent was waived due to the study's retrospective design using de-identified pre-existing public data.

### Definitions

Complicated phenotype was defined as any of the following: (1) hemodynamic instability suggested by the usage of intravenous vasopressors or major organ support including mechanical ventilation, continuous renal replacement therapy, and intra-aortic balloon pump and extra-corporeal membrane oxygenation; or (2) the event of cardiopulmonary resuscitation during the index hospitalization. All other cases were classified as being a non-complicated phenotype.

Comorbidities were defined as follows: hypertension as ≥5 billing records of codes (I10–I13 and I15); diabetes mellitus as ≥5 billing records of codes (E11–E14); dyslipidemia as ≥5 billing records of codes (E78); stroke as codes (I60–I64) with hospital admission ≥3 days and either computed tomographic or magnetic resonance imaging of brain; connective tissue disease (CTD) as codes of autoimmune systemic diseases (M32–M36, L93–L94, and N085); malignant diseases as hospital admission ≥3 days with ICD codes (I60–I64); or a code of the special national program for rare intractable diseases (V193).

### Outcomes

The annual incidence of suspected acute myocarditis was calculated with reference to the national population of the resident registration in 2009 [26]. Annual mortality data were obtained from Microdata Integrated Service managed by Statistics Korea [27]. The primary endpoint was the all-cause deaths at 30 days and 10 years after suspected acute myocarditis hospitalization. The secondary endpoint was cardiovascular deaths at the same time points. The other secondary endpoint was the excessive mortality driven by index hospitalization.

### Statistical analysis

Numerical data are provided as mean (SD) or median (IQR), and categorical variables as frequencies (percentages). The difference by age groups was evaluated using the two-way chi-square test, Fisher's exact test, and ANOVA. The cumulative survival rate of all-cause death and cardiovascular death according to age group was compared using the Kaplan-Meier curves and the log-rank tests. The Cox proportional hazard regression analyses were used to predict the hazard ratio (HR), with a confidence interval (CI) of 95%. All HRs were adjusted for age and sex. The excess mortality rate was calculated to investigate the difference between the mortality of the study population and that of the general population, using the flexible parametric survival models which allowed the incorporation of time-dependent effects and utilized the restricted cubic spline function to estimate the cumulative hazard function [28]. Statistical significance was assessed as a two-sided p-value $<0.05$. Statistical analyses were performed using SAS version 9.4 (SAS Institute, Cary, NC, USA) and Stata/MP version 16.0 (Stata Corporation, College Station, TX, USA).

## Results

### The national epidemic of clinically suspected acute myocarditis in Korea

The final study population comprised 2,988 patients (Fig 1). The mean age at presentation was 51.0 ± 16.9 years, and women were 1,379 (46.2%). The incidence rate of suspected acute de novo myocarditis was 0.57 per 100,000 people of all adults aged 20–79. Annual prevalence did not change over the study period from 2006 to 2018 (S1 Table). The baseline characteristics according to differential age groups are shown in Table 1. The median duration of hospital admission was 14 (8–22) days. The all-cause mortality rates at 30 days and 10 years were 6.6% (197 patients) and 25.6% (762 patients), respectively (Fig 2). A total of 362 (12.1%) patients represented the complicated phenotype, which was a significant factor in predicting poor clinical outcomes (S1 Fig). Although our data provided an indirect approximation to acute myocarditis, the long-term mortality was consistent with the finding of other literature (Fig 3) [7, 8, 10, 11, 13–15, 17, 19–22].

### Clinically suspected acute myocarditis in young adults (ages 20–39)

A total of 848 patients (28.4%) were classified as being in the young-adult age group. This age group comprised more men (63%) than women, and 499 (58.8%) patients of this group were

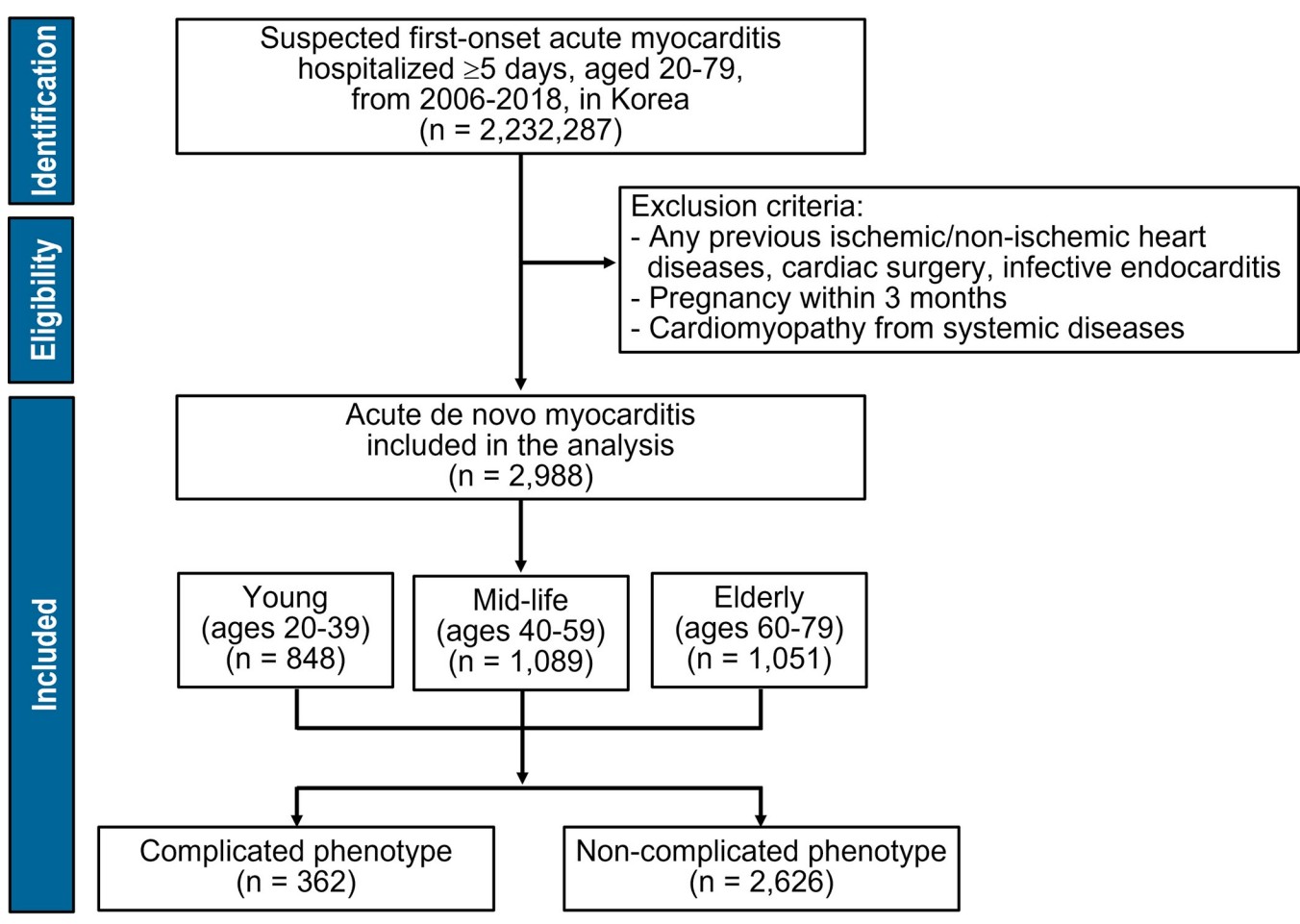

**Fig 1. Flow chart of the study population.**

**Table 1. Baseline characteristics.**

| | Young adult (Ages 20–39) (n = 848) | Mid-life adult (Ages 40–59) (n = 1,089) | Older adult (Ages 60–79) (n = 1,051) | P-value |
|---|---|---|---|---|
| Age (years) | 29.8 ± 5.9 | 49.4 ± 5.5 | 69.8 ± 5.6 | <0.001 |
| Male (%) | 536 (63.2) | 581 (53.4) | 492 (46.8) | <0.001 |
| Body mass index (kg/m$^2$) | 23.9 ± 4.1 | 24.1 ± 3.5 | 23.6 ± 3.6 | <0.001 |
| Admission via ED (%) | 499 (58.8) | 480 (44.1) | 445 (42.3) | <0.001 |
| Hospital stay (days) | 13 (6–19) | 15 (8–22) | 15 (9–24) | <0.001 |
| **Comorbidities** | | | | |
| Hypertension (%) | 28 (3.3) | 241 (22.1) | 556 (52.9) | <0.001 |
| Diabetes (%) | 20 (2.4) | 156 (14.3) | 306 (29.1) | <0.001 |
| Prior stroke (%) | 2 (0.2) | 33 (3.0) | 112 (10.7) | <0.001 |
| Connective tissue disease (%) | 35 (4.1) | 44 (4.0) | 35 (3.3) | 0.592 |
| Malignancy (%) | 10 (1.2) | 68 (6.2) | 117 (11.1) | <0.001 |
| **Complicated phenotype (%)**[*] | 92 (10.8) | 118 (10.8) | 152 (14.5) | 0.378 |
| Mechanical ventilation (%) | 58 (6.8) | 72 (6.6) | 101 (9.6) | 0.018 |
| CRRT (%) | 5 (0.9) | 9 (0.8) | 10 (1.0) | 0.676 |
| IABP (%) | 10 (1.2) | 14 (1.3) | 13 (1.2) | 0.978 |
| ECMO (%) | 13 (1.5) | 12 (1.1) | 4 (0.4) | 0.034 |
| Cardiopulmonary resuscitation (%) | 41 (4.8) | 43 (3.9) | 39 (3.7) | 0.444 |
| **Medical treatment** | | | | |
| Norepinephrine (%) | 62 (7.3) | 74 (6.8) | 80 (7.6) | 0.762 |
| Inotropics (%) | 66 (7.8) | 62 (5.7) | 53 (5.0) | 0.037 |
| Intravenous nitrate (%) | 149 (17.6) | 176 (16.2) | 133 (12.7) | 0.008 |
| Loop diuretics (%) | 261 (30.9) | 302 (27.7) | 299 (28.4) | 0.320 |
| Antiarrhythmics (%) | 32 (3.8) | 39 (3.6) | 43 (4.1) | 0.825 |
| ACEI/ARB (%) | 244 (28.8) | 254 (23.3) | 260 (23.9) | 0.020 |
| MRA (%) | 127 (15.0) | 133 (12.2) | 109 (10.0) | 0.010 |
| Beta-blockers (%) | 201 (23.7) | 221 (20.3) | 167 (15.3) | <0.001 |
| Digoxin (%) | 24 (2.8) | 35 (3.2) | 47 (4.3) | 0.120 |

[*]The same patient was involved in ≥2 treatments.

ACEI, angiotensin converting enzyme inhibitor; AP, arterial pressure; ARB, angiotensin receptor blocker; CRRT, continuous renal replacement therapy; ECMO, extra-corporeal membrane oxygenation; ED, emergency department; IABP, intra-aortic balloon pulsation; MRA, mineralocorticoid receptor antagonist; PAP, pulmonary arterial pressure

hospitalized via the emergency department. The complicated phenotype was identified in 92 (10.8%) patients. Of those with the complicated phenotype, 41 underwent cardiopulmonary resuscitation, and 25 died during the index hospitalization. At 30 days, 40 (4.7%) patients had died from any cause, which was independently associated with the complicated phenotype (HR: 13.92, 95% CI: 8.43–22.97), underlying diabetes (HR: 3.99, 95% CI: 1.79–8.92), CTD (HR: 2.90, 95% CI: 1.25–6.74), and malignant diseases (HR: 7.64, 95% CI: 3.04–19.17) (S2 Fig). Most short-term deaths occurred within the first 30 days, and the risk of early death significantly decreased thereafter (Fig 2).

### Clinically suspected acute myocarditis in mid-life (ages 40–59)

There were 1,089 patients in the mid-life adult group. Compared to their younger peers, more patients in this group had hypertension, diabetes, previous stroke, and malignant diseases

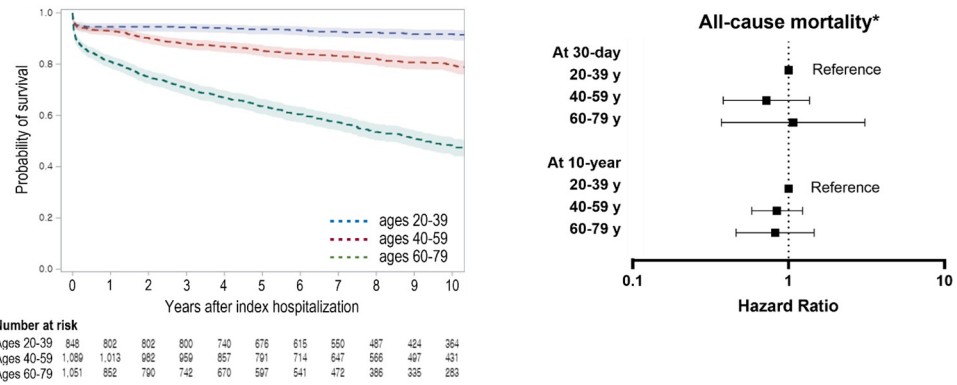

**Fig 2. All-cause mortality according to differential age groups.** *Adjusted for age and sex.

(Table 1). The complicated cases were 118 (10.8%). At 30 days, 52 (4.8%) patients had died, and the risk factors were similar to those in the older-adult group independently associated with comorbidities and complicated presentation (S2 Fig).

## Clinically suspected acute myocarditis in older adults (ages 60–79)

There were 1,051 patients in the older-adult group, and this group comprised fewer men (46.8%) than women. A large number of patients had underlying comorbidities, including malignancy in 117 (11.1%). There were 152 (14.5%) complicated cases. At 30 days, the all-cause mortality was 10.0% (105 patients), which was associated with age (HR: 1.07, 95% CI: 1.06–1.09), male sex (HR: 1.41, 95% CI: 1.18–1.69), underlying hypertension (HR: 1.31, 95% CI: 1.10–1.55), diabetes (HR: 1.31, 95% CI: 1.1–1.57), previous stroke (HR: 1.56, 95% CI: 1.23–1.98), malignancy (HR: 2.84, 95% CI: 2.23–3.62), and complicated phenotype (HR: 3.10, 95% CI: 2.29–4.03) (S2 Fig). During the 10 years of follow-up, the majority of deaths were attributed to non-cardiac causes (S3 Fig).

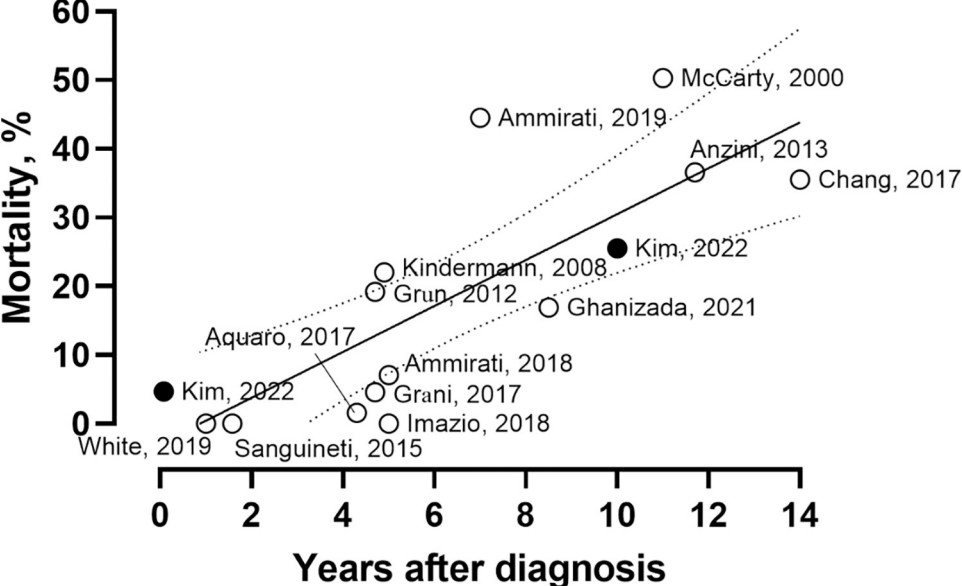

**Fig 3. Comparison of the all-cause mortality results with the previous studies** [7, 8, 10, 11, 13–15, 17, 19–22].

### Excess mortality associated with the index hospitalization

The long-term excess mortality associated with the clinically suspected acute myocarditis in the young adult group was 254 per 1,000 person-years over a 10-year follow-up, which was higher compared to that of the age and sex-matched general population (Fig 4). Conversely, the 10-year excess mortalities in the mid-life and older adult groups were not as high as those of the young adult group (Fig 4). Although the absolute acute and long-term mortality rates increased with age, the adverse impact of suspected acute myocarditis was much worse in the young adult group.

## Discussion

In the present study, we showed that (1) all-cause mortality in patients with clinically suspected acute myocarditis was highly dependent on patients' age; (2) the older adults were at the highest risk of early death, particularly in those with comorbidities or complicated presentation; and (3) even the young patients without complicate phenotype had significant risk in long-term excess mortality associated with clinically suspected acute myocarditis. The strength of this study was that it provided a national approximation of long-term mortality data based on age. In addition, our study includes 46% of female patients, which is the highest proportion in the literature to date.

Most patients with acute myocarditis experience an uncomplicated and self-limited course, and their subsequent sequelae are considered negligible after the acute phase has ended [20–22]. Some patients, however, experience serious clinical courses and unfavorable long-term cardiac outcomes via progressive LV dysfunction even after temporary recovery [9, 10, 14, 15, 17]. Such progressive LV functional deterioration has even been observed in cases with uncomplicated cases [9, 12, 15]. In the present study, patients without complicated phenotypes were involved one-third of the 30-day deaths and long-term excess mortality. The long-term residual risk associated with acute myocarditis was also identified in the Danish national cohort [19]. In this Danish study, all-cause mortality was 16.9% over 8.5 years, and even younger patients who recovered from acute myocarditis without complications were associated with an excessive risk of heart failure and death compared to age and sex-matched controls in the long term.

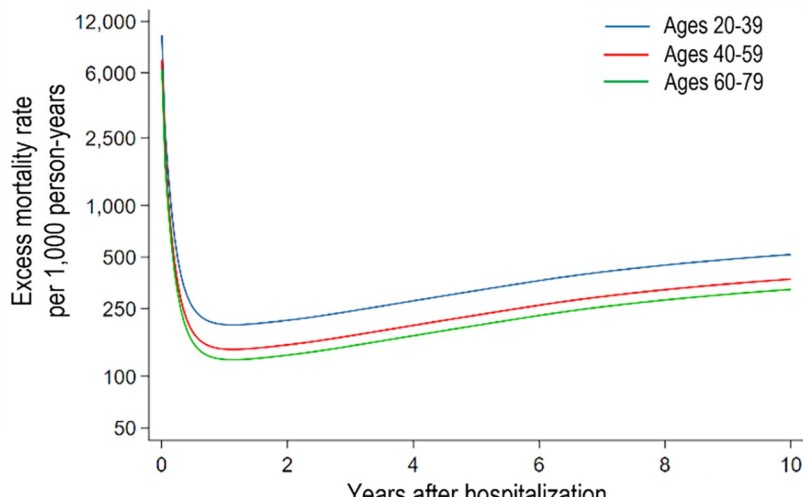

| Excess mortality per 1,000 person-years | |
| --- | --- |
| **At 30-day** | n |
| 20-39 y | 1442 |
| 40-59 y | 991 |
| 60-79 y | 862 |
| **At 10-year** | |
| 20-39 y | 254 |
| 40-59 y | 175 |
| 60-79 y | 152 |

**Fig 4. Excess mortality compared to the age and sex-matched general population associated with the index hospitalization in differential age groups.**

Although many acute myocarditis studies have focused on adolescents and young men, the spectrum of patients with acute myocarditis is much wider. In a Finnish study, the highest acute myocarditis occurrence age for men was 16–20 years, while that for women was 56–60 years [11]. In our study, age was a major determinant of mortality in patients with suspected acute myocarditis. In addition, the older adult group had a different distribution of underlying diseases compared to the young adult group. The clinically suspected myocarditis associated with CTD is likely to be more frequent in young adults, and CTD was an independent predictor for short-term mortality in the young adult group in our study, which was consistent with the literature [29]. In older adults, myocarditis might be more frequently associated with other causes than in young adults. For example, dilated cardiomyopathy is often triggered by acute viral infection mimicking acute myocarditis [30, 31].

Our study examined acute myocarditis before the start of the COVID-19 pandemic. The COVID-19 pandemic has proven again that older adults are clearly vulnerable to viral myocarditis and viral infection-related cardiac injury. If SARS-CoV2-related myocarditis has a similar natural history of acute myocarditis as that associated with one of the previously known cardiotropic viruses, many post-myocarditis people might be at risk of adverse outcomes within the next 10 years.

The present study has several limitations. First, this was a retrospective observational study based on information used for insurance claims. Because the entire Korean population is covered by national insurance, most medical activities have been recorded in the NHIS database. However, some procedures such as cardiac magnetic resonance imaging beyond the scope of insurance coverage depending on the time of index hospitalization were not included in the analyzed data. Second, we accepted the diagnostic ICD codes of acute myocarditis officially reported to the governmental insurance agencies to enroll the study population. From the reported number of invasive coronary angiography and endomyocardial biopsy, 652 (21.8%) and 31 (1.0%), respectively, the diagnosis of acute myocarditis was likely to be made by excluding alternative diseases in many cases. This was why we used the term "clinically suspected acute myocarditis" instead of "acute myocarditis". To mitigate this concern, we employed a careful operational definition for clinically suspected acute myocarditis and a wide range of pre-specified exclusion criteria. Although our data suggested an indirect approximation to acute myocarditis, the long-term mortality, consistent with the finding of other literature, supported that this approach was not deceptive.

## Conclusion

In conclusion, for hospitalized patients with clinically suspected acute myocarditis, the short-term mortality is high both in young and older adults. High-risk features include advanced age, underlying systemic comorbidities, and serious clinical presentation. Excess mortality remains significant for up to 10 years after the index hospitalization in young adults. It is time to discuss a specific management strategy for the post-myocarditis population against their long-term health threats.

## Supporting information

**S1 Fig.** All-cause (upper panel) and cardiovascular (lower panel) mortality according to differential clinical severity.
(TIF)

**S2 Fig. Risk factors associated with all-cause death at 30 days in differential age groups.**
(TIF)

**S3 Fig. Cardiovascular mortality according to differential age groups.**
(TIF)

**S1 Table. The annual incidence of suspected acute myocarditis during the study period.**
(DOCX)

## Author Contributions

**Conceptualization:** Hae Ok Jung.

**Formal analysis:** Hoseob Kim, Yoonjong Bae.

**Resources:** Hae Ok Jung.

**Supervision:** Doo Soo Jeon.

**Writing – original draft:** Mi-Jeong Kim.

**Writing – review & editing:** So Young Lee.

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
