## [Decision Letter · Decision Letter 0]

3 Nov 2022

PONE-D-22-2327710- year survival outcome after de novo hospitalized acute myocarditis in Korean adults: a nationwide study in the pre-COVID-19 eraPLOS ONE

Dear Dr. JUNG,

Thank you for submitting your manuscript to PLOS ONE. After careful consideration, we feel that it has merit but does not fully meet PLOS ONE’s publication criteria as it currently stands. Therefore, we invite you to submit a revised version of the manuscript that addresses the points raised during the review process.

We look forward to receiving your revised manuscript.

Kind regards,

Joshua M. Hare, M.D., F.A.C.C., F.A.H.A.

Academic Editor

PLOS ONE

Journal Requirements: 

Additional Editor Comments (if provided):

The authors need to include the details/methodology for diagnosing patients with myocarditis. All Figures and Tables should be presented initially in the Results section not in the Discussion.

Reviewers' comments:

Reviewer's Responses to Questions

**Comments to the Author**

1. Is the manuscript technically sound, and do the data support the conclusions?

Reviewer #1: No

2. Has the statistical analysis been performed appropriately and rigorously? 

Reviewer #1: N/A

3. Have the authors made all data underlying the findings in their manuscript fully available?

Reviewer #1: No

4. Is the manuscript presented in an intelligible fashion and written in standard English?

Reviewer #1: No

5. Review Comments to the Author

Reviewer #1: A crucial information that is missing is how the patients were diagnosed with myocarditis. It is important to know, how many patients underwent endomyocardial biopsy and how many CMR to get an idea, how accurate the diagnosis of myocarditis was.

6. PLOS authors have the option to publish the peer review history of their article (what does this mean?). If published, this will include your full peer review and any attached files.

Reviewer #1: No

---

## [Author Response · Author response to Decision Letter 0]

16 Dec 2022

Please refer to the response letter uploaded in the system.

---

## [Decision Letter · Decision Letter 1]

20 Jan 2023

10-year survival outcome after clinically suspected acute myocarditis in adults: a nationwide study in the pre-COVID-19 era

PONE-D-22-23277R1

Dear Dr. JUNG,

We’re pleased to inform you that your manuscript has been judged scientifically suitable for publication and will be formally accepted for publication once it meets all outstanding technical requirements.

Kind regards,

Joshua M. Hare, M.D., F.A.C.C., F.A.H.A.

Academic Editor

PLOS ONE

Additional Editor Comments (optional):

Two edits were requested by the Reviewer, who otherwise considered that all comments were addressed

Reviewers' comments:

Reviewer's Responses to Questions

**Comments to the Author**

1. If the authors have adequately addressed your comments raised in a previous round of review and you feel that this manuscript is now acceptable for publication, you may indicate that here to bypass the “Comments to the Author” section, enter your conflict of interest statement in the “Confidential to Editor” section, and submit your "Accept" recommendation.

Reviewer #1: All comments have been addressed

2. Is the manuscript technically sound, and do the data support the conclusions?

Reviewer #1: Yes

3. Has the statistical analysis been performed appropriately and rigorously? 

Reviewer #1: Yes

4. Have the authors made all data underlying the findings in their manuscript fully available?

Reviewer #1: Yes

5. Is the manuscript presented in an intelligible fashion and written in standard English?

Reviewer #1: (No Response)

6. Review Comments to the Author

Reviewer #1: The authors have carefully adressed my comments. I only have 2 minor comments that still have to be addressed:

1) Sentence: The strength of this study was that it provided a national approximation of long-term mortalities by differential age

I would recommend to write "..long-term mortality data based on age."

2) In addition, our study includes 46% of female patients, which is the highest proportion in the literature to date.

Although our data suggested an indirect approximation to acute myocarditis, the long-term mortality consistent with the finding of other literature supported that this approach was not deceptive.

I think the word "was" is missing in the second sentence

7. PLOS authors have the option to publish the peer review history of their article (what does this mean?). If published, this will include your full peer review and any attached files.

Reviewer #1: No

---

## [Editor Report · Acceptance letter]

23 Jan 2023

PONE-D-22-23277R1 

10-year survival outcome after clinically suspected acute myocarditis in adults: a nationwide study in the pre-COVID-19 era 

Dear Dr. Jung:

I'm pleased to inform you that your manuscript has been deemed suitable for publication in PLOS ONE. Congratulations! Your manuscript is now with our production department. 

Kind regards, 

on behalf of

Dr. Joshua M. Hare 

Academic Editor

PLOS ONE